# A Short Form of a Questionnaire on Attitudes Toward Patient Communication in Nurses and Nursing Students (ACO-R): Assessment of Psychometric Properties

**DOI:** 10.3390/healthcare12242546

**Published:** 2024-12-17

**Authors:** Alicia Tamarit, María del Carmen Giménez-Espert, Laura Lacomba-Trejo, Vicente Prado-Gascó

**Affiliations:** 1Social Psychology Department, Faculty of Psychology and Speech Therapy, Universitat de València, 46010 Valencia, Spain; alicia.tamarit@uv.es (A.T.); vicente.prado@uv.es (V.P.-G.); 2Department of Nursing, Faculty of Nursing and Chiropody, Universitat de València, 46010 Valencia, Spain; 3Developmental and Educational Psychology Department, Faculty of Psychology and Speech Therapy, Universitat de València, 46010 Valencia, Spain; laura.lacomba@uv.es

**Keywords:** ACO-R, nursing, attitudes toward patient communication, validation

## Abstract

**Background:** Attitudes toward the communication of nurses and nursing students with patients can influence the health outcomes of their patients. The present study aimed to develop and validate an abbreviated form of the Attitudes Toward Communication (ACO) scale for nurses and nursing students. **Methods**: Two types of participants were included in this study, 385 nurses and 1408 nursing students (67.30% of the nurses and 82.10% of the students were women). After obtaining their informed consent. Attitudes toward communication (ACO) of nurses and nursing students with patients were evaluated. **Results:** Internal consistency and construct validity analyses were conducted. A 12-item instrument (ACO-R) was obtained that maintained the factorial structure and ensured the homogeneous distribution of items in the different subscales. The same solution was found for both samples. **Conclusions**: The instrument showed adequate internal consistency and validity indices. The ACO-R instrument is an efficient, valid, and reliable measure to assess ACO among nursing students and nurses.

## 1. Introduction

Communication between nurses and patients is an exchange of information whereby nurses must have a holistic view of the person. In this way, nurses can learn about patients’ needs, establish a therapeutic relationship, and increase satisfaction with care [1]. The communicative process may be influenced by nurses’ attitudes [2,3], these attitudes being facilitating factors or barriers towards their communication with the patient [4]. The biopsychosocial model [5] highlights that patient-centered communication is essential in achieving better adherence to treatment, better assessment of services, better quality of life, and better survival outcomes [6].

From a theoretical framework, the Theory of Reasoned Action [7] and its extension, the Theory of Planned Behavior [8], emphasize the central role of attitudes, subjective norms, and perceived behavioral control in predicting intentions and behaviors. According to these theories, attitudes are composed of the following three key components: affective, behavioral, and cognitive. The affective component refers to the emotional responses or feelings, such as a positive disposition toward engaging in nurse–patient communication. The behavioral component addresses the tendency or intention to act in a certain way, such as actively striving to communicate effectively with patients. Finally, the cognitive component involves beliefs or knowledge about patient behavior, such as acknowledging the importance of effective communication in improving patient outcomes. In the context of nurse–patient communication, these components interact, facilitating attitudes that increase the willingness to communicate more effectively. The theories applied to nurse–patient communication suggest that by fostering favorable attitudes toward communication, nurses can adopt a patient-centered approach, promoting more effective interactions and ultimately improving health outcomes.

Consequently, communication is one of the most important skills in the work of nurses [9]. Positive attitudes towards it have positive repercussions on nurses but also on patients [10,11]. In this sense, a positive communicative attitude reduces healthcare costs, improves teamwork, increases job satisfaction, and reduces the risk of burnout [12]. There is some research indicating there might be some gender differences—women tend to present higher communication skills and higher emotional intelligence—but there is not a consensus in the literature, especially regarding a female-dominated profession such as nursing [13]. As for the people being treated, good communication increases informed decision-making, autonomy, and adherence to treatment and decreases healthcare costs [14]. In line with the above, its absence increases the unnecessary use of health services, the waste of health resources, and puts patients at unnecessary risk, potentially even reducing their survival [15,16].

Therefore, it is considered essential for nurses and nursing students to acquire positive communication attitudes towards patients. Despite the above, in the Spanish context, there are a few tools that evaluate communication in clinical nursing practice without considering oncology patients or nursing services [17]. The literature is even scarcer on instruments that have been validated for both practicing nurses and nursing students [9,18]. For this reason, in 2018, the Attitudes to Patient Communication (ACO) scale (registration number: UV-MET-201917R) was developed [4], which was later adapted and validated for use with nursing students in 2021 [19]. The ACO scale [4] was originally developed to assess the attitudinal dimensions of communication among nursing professionals, with the aim of fostering better interpersonal interactions in healthcare settings. The scale’s development followed rigorous psychometric procedures, including a thorough literature review to identify relevant constructs, the development of an item pool, the expert validation of items, and iterative testing across diverse samples. The ACO assesses key dimensions of nurse–patient communication, such as empathy, clarity, and emotional regulation, which are critical for effective healthcare delivery and patient satisfaction.

The scale has been shown to be a useful, reliable, and valid measure in multiple studies [20,21,22]. This instrument consists of 25 items divided into three sub-scales (cognitive, affective, and behavioral factors). Furthermore, the ACO has been validated against theoretically relevant constructs like social skills, empathy, and emotional intelligence, reinforcing its utility as a multidimensional measure of communication attitudes [21,22].

The instrument has been successfully used with samples of professional nurses and samples of nursing students [4,20]. In all cases, the instrument has shown strong psychometric properties in terms of both reliability and validity. The original 2018 study yielded adequate goodness-of-fit indices for the 25-item structure, for reliability indices (α = 0.76), EFA (RMSEA = 0.42; GFI = 0.99), and CFA (S-B *χ*^2^ = 525.09; *df* = 272; *p <* 0.05; CFI = 0.91; IFI = 0.91) [4]. Discriminant validity was assessed by verifying that all correlations between the various dimensions were below 0.85 [23]. Additionally, the square root of the Average Variance Extracted (AVE) was required to exceed the correlation between pairs of dimensions [24], with all dimensions meeting this criterion. These findings highlight the robust psychometric properties of the instrument, supporting its use as a reliable and valid measure of communication attitudes in nursing.

These indices were also proven to be adequate in the 2021 version, which was adapted for nursing students, for reliability indices (α = 0.84), EFA (RMSR = 0.03; GFI = 0.99), and CFA (*χ*^2^ = 2347.59; *df* = 272; *p <* 0.05; CFI = 0.92; IFI = 0.92) [19]. Moreover, it has been observed that the ACO is related to greater social skills, empathy, and emotional intelligence, among other factors [21,22]. Research on ACO consistently confirms its adequacy as a reliable and useful tool for the assessment of attitudes towards communication in nurses [20].

Despite these good indications, there is a growing need to reduce the length of questionnaires to make measurement faster, simpler, more reliable, and more productive [25]. Shorter instruments allow us to achieve a higher response rate and maintain attention and motivation towards the answers [26]. For this reason, taking into account the previous good results of the ACO questionnaire and the need to economize the time and effort used in the evaluations, this study aimed to develop an abbreviated form of the scale and evaluate its psychometric properties in a sample of nursing students and nurses. Therefore, the main objective of this study was to develop a short version of the ACO scale (ACO-R) and analyze its psychometric properties in both nurses and nursing students. For this purpose, five specific objectives have been formulated as follows: (1) examine the psychometric properties of the ACO-R scale by evaluating its reliability through internal consistency (Cronbach’s alpha) and test–retest reliability, assessing validity through factorial validity (confirmatory factor analysis) and discriminant and convergent validity; (2) analyze the mean differences in ACO-R scores across professional profiles (nursing students versus nurses) and gender (women versus men); and (3) provide comparative criteria for various subgroups within the sample. This comprehensive approach ensures a thorough psychometric evaluation is carried out, and it facilitates meaningful subgroup comparisons across nursing contexts.

## 2. Materials and Methods

### 2.1. Participants

A total of 1408 nursing students and 385 nurses participated in the study. In this regard, 82.10% of the nursing students were women. Their ages ranged from 17 to 55 years (*M* = 21.79; *SD* = 5.34). Most of them, 1004 students (71.30%), were unemployed, while 159 (11.30%) had temporary contracts and 109 (7.70%) held permanent positions. As for their academic course, 29.6% of them were in the first year of their degree, 25% on the second year, 23.8% on the third, and 21.6% on their fourth year. As for the nurses in the sample, 67.30% of them were women between the ages of 23 and 64 years (*M* = 45.46; *SD* = 10.91). Their employment status was divided proportionally between temporary workers, of which 56 (14.50%) had a temporary contract and 102 (26.50%) were temporary workers, and those with a permanent position, which included 185 (48.10%) nurses. Their years of experience ranged from 1 month to over 43 years (*M* = 21.64; *SD* = 20.50). They worked mainly in the Emergency Department (n = 41, 10.6%), Intensive Care Unit (ICU) (n = 37, 9.6%), Surgery (n = 28, 7.3%), and Internal Medicine (n = 26, 6.8%), but they also worked in many other sections, such as Digestive, Pediatrics, Neonatology, Hemodialysis, Trauma, Pulmonology, Intensive Respiratory Rehabilitation Unit (REA), Operating Room, Cardiology, Comprehensive Medicine, and Oncology–Hematology–Cardiology departments, among others.

### 2.2. Measure

#### Attitudes Towards Communication

The Attitudes Toward Communication in Nurses (ACO) questionnaire was administered [4] (intellectual property registered at the University of Valencia on 8 April 2019, registration number: UV-MET-201917R). A language adaptation was made in the case of nursing students [19]. The initial questionnaire had 25 items divided into three dimensions, affective, cognitive and behavioral. It had a Likert-type response format ranging from 1 = strongly disagree to 5 = strongly agree. In previous studies, the instrument had shown adequate psychometric properties in a sample of nurses and nursing students [4,20].

### 2.3. Procedure and Study Design

The sample was collected between 2015 and 2019. Nursing undergraduate students from five universities in the Valencian Community (Spain) participated. Of these, three were private and two were public. The inclusion criteria for students were that participants were actively enrolled in an official nursing degree and that they consented to their anonymous and confidential participation in the study.

In Spain, the nursing degree program spans four academic years, which totals 240 credits under the European Credit Transfer System (ECTS), with students completing 60 ECTS credits per year. For scale, each ECTS credit corresponds to approximately 30 h of academic work. Once this program finishes, graduates receive a nursing degree that qualifies them to practice professionally in Spain as well as in other countries within the European Union.

As for the sample of nurses, it was collected from six hospitals in the Valencian Community (Spain). The inclusion criteria for nurses were that they were actively employed, whether by temporary or permanent contracts, and that they consented to their anonymous and confidential participation in the study. Therefore, the study involved a single evaluation through a cross-sectional design.

All participants signed informed consent forms. This study was authorized by their respective ECCRs (Ethics Committees for Clinical Research) and the Research Ethics Committee of the University where the study was performed (H1529396558647). After this, participants were given three weeks to complete the questionnaire, which lasted around 10 min, and were provided with a collection box at each unit to deposit their confidential answers. These collection boxes were used to ensure the answers were anonymous and to prevent any possible identification of the participant. In order to improve response rates, reminder emails were sent two weeks after the initial distribution, and the completed questionnaires were retrieved after three weeks.

### 2.4. Data Analysis

Data analysis was carried out using SPSS version 27.0 (IBM Corporation, Armonk, NY, USA) for the descriptive analyses, calculation of internal consistency, *t*-test, Pearson correlations and comparison criteria.

In the first instance, reliability analyses were conducted. For Average Variance Extracted (AVE), scores greater than 0.50 were considered adequate; as for Cronbach’s Alpha (α) and Composite Reliability (CR), values over 0.70 were considered adequate [27].

Then, a Confirmatory Factor Analysis (CFA) was performed using Structural Equation Modeling software (EQS, version 6.3, Multivariate Software, Inc., Los Angeles, CA, USA). To validate the factor structure of the sub-scales, the CFA was carried out, and the Satorra–Bentler maximum likelihood goodness-of-fit (ML) indices were taken into account. The adequacy of the CFAs was tested by the significance of the chi-square and the robust Satorra–Bentler correction (S-B *χ*^2^) [28]. In addition to the above, the adequacy of the goodness-of-fit indices of the models was assessed using the Comparative Fit Index (CFI) and the Incremental Fit Fixation (IFI), whose values ≥ 0.90 were considered adequate [29]. Finally, the Root Mean-Square Error of Approximation (RMSEA) was sampled, whose scores should be ≤0.08 [30].

Furthermore, *t*-test analyses were conducted in order to examine the mean differences in attitudes toward patient communication across professional profiles (students and nurses), and gender (women and men). Pearson correlations were conducted in order to study the intercorrelations between the dimensions for ACO-R and ACO-25 for both students and nurses. Percentiles from 10 to 90 were calculated for the following 6 different subsamples: students, nurses, female students, female nurses, male students, and male nurse; these are the comparison criteria.

## 3. Results

### 3.1. Item Analysis and Reliability

Reliability analyses were conducted on the short form of the ACO-R instrument, shown in Table 1. Reliability indices suggested keeping all items, as all of the alpha indices of their respective factors decreased when eliminating that item in both nursing students and nurses (Table 1). All dimensions presented acceptable internal consistency coefficients.

### 3.2. Validity Analysis of the Instrument

The initial instrument (ACO) consisted of 25 items divided into three dimensions (affective, behavioral, and cognitive). Through content analysis, the study of the factor loadings of the items, and by taking into account the objective of finding a comparable final solution in both samples (nurses and nursing students), 12 items that maintain the initial structure of the questionnaire (affective, behavioral and cognitive) and that are distributed among the subscales in a comparable manner were selected. A content validation process was conducted with more than five experts, as recommended in the literature [31]. The panel was composed of 10 professionals with advanced academic and practical expertise in their respective fields to guarantee a high level of proficiency and uniformity in knowledge. The panel included five nursing researchers, each with a minimum of 10 years of clinical experience and at least a master’s degree in their field. This criterion minimized variability in educational background and ensured a consistent knowledge base among the participants. Additionally, three psychometricians and two psychologists, recognized for their expertise in scale development and evaluation, contributed their insights to refine the clarity and relevance of the scale items. The experts were provided with the online tool and were given three weeks to complete their evaluations. This timeframe allowed them to thoroughly review the items and provide detailed feedback on their relevance, clarity, and overall alignment with the objectives of the study. This involved assessing each item’s representativeness and its alignment with the content intended for measurement. Moreover, they checked for the comprehensive coverage of all aspects of the focal variables, avoiding redundancy. After removing 7 items, 12 remained for further validation.

These items were (F1) items 3, 5, 6, 7, (F2) 14, 15, 16, 18, and (F3) items 22, 23, 24, and 25. After analyzing the reliability of the items, the internal validity of the instrument was assessed using CFA (Table 2). The goodness-of-fit indicators for the three-factor solution in the 12-item version were satisfactory (Table 2). Additionally, and as the score computation suggested, a second-order model was tested to examine whether a higher-order factor existed. Results showed that, for both students and nurses, a second-order model provided a marginally better fit for the data structure. This second-order model yielded a slightly higher *χ*^2^, a slightly lower Satorra–Bentler *χ*^2^, and, in the case of nurses, a slightly narrower confidence interval and higher IFI and CFI scores. These results indicate that a higher-order factor exists together with the three-factor structure previously observed. Furthermore, the indices for ACO-R were more adequate than those for ACO-25, suggesting the short form of the ACO questionnaire is psychometrically sound. This was true for both the three-factor and second-order model calculated, which presented similar results.

In order to test whether the ACO-R was invariant across the groups (students and professionals), measurement invariance analyses were conducted. As recommended by [32], the evaluation of measurement invariance was conducted to ensure the comparability of the latent constructs across groups.

First, separate confirmatory factor analyses (CFA) were conducted for each group, then a multigroup analysis was performed including both samples, taking into account the fit indices (Table 3). The single-group solutions indicated acceptable fit indices for students (*χ*^2^ = 215.23, *df* = 51; RMSEA = 0.03 [90% CI: 0.020–0.035]; IFI = 0.98; CFI = 0.98) and professionals (*χ*^2^ = 277.03, *df* = 51; RMSEA = 0.06 [90% CI: 0.044–0.71]; IFI = 0.93; CFI = 0.93). The same was obtained in the case of the multigroup analysis (Equal Form: *χ*^2^ = 404.56, *df* = 102; RMSEA = 0.06 [90% CI: 0.054–0.066]; IFI = 0.98; CFI = 0.98). Based on these fit indices, it was possible to establish configural invariance and invariance in the factor structure (equal form). The next step involved examining whether the factor loadings for each indicator were equivalent across the two groups. To achieve this, constraints were applied in the multigroup model to equalize the contribution of each item to its respective factor. The chi-square difference was analyzed between the unconstrained model and the model with constrained loadings. If a significant difference was observed (Table 4), full metric invariance could not be assumed.

However, as suggested by Aldás-Manzano (2013) [32], partial metric invariance can be assumed if at least two indicators per factor exhibit equivalent loadings (partial equal factor loadings) or if at least two intercepts are equivalent (partial equal intercepts). To this end, in the model with constrained loadings, a Lagrange multiplier test was conducted to determine whether releasing specific constraints would lead to a significant change in the chi-square values (Table 4).

Upon analyzing the significance of chi-squared changes when releasing each constraint, the Lagrange multiplier test revealed issues with items ACO6 and ACO5 (from the affective factor), ACO15 and ACO18 (from the behavioral factor), and ACO23 (from the cognitive factor); therefore, at least two indicators per factor were maintained. In conclusion, the results indicate that while full metric invariance was not achieved, partial metric invariance was supported, allowing the meaningful comparison of the latent constructs across groups.

With the aim to explore in depth the differences between ACO-R and ACO-25 for each group, *t*-test analyses were calculated. These analyses were conducted to examine the differences in attitudes toward patient communication between professional profiles (students and nurses), and gender (women and men) (Table 5).

The mean differences between students and nurses showed significant differences. Students presented higher behavioral (*M*_Student_ = 4.56; *SD*_Student_ = 0.74; *M*_Nurses_ = 4.08; *SD*_Nurses_ = 0.90; *t*_505.28_ = 9.27; *p* < 0.001) and cognitive scores (*M*_Student_ = 4.72; *SD*_Student_ = 0.64; *M*_Nurses_ = 4.41; *SD*_Nurses_ = 0.88; *t*_473.16_ = 6.49; *p* < 0.001), as well as total attitude scores (*M*_Student_ = 4.54; *SD*_Student_ = 0.57; *M*_Nurses_ = 4.29; *SD*_Nurses_ = 0.79; *t*_493.44_ = 5.74; *p* < 0.001). The size effect of these differences ranged from 0.62 to 0.78. Gender differences were also observed. Women showed higher behavioral (*M*_Women_ = 4.51; *SD*_Women_ = 0.79; *M*_Men_ = 4.27; *SD*_Men_ = 0.84; *t*_436.51_ = 4.56; *p* < 0.001), cognitive (*M*_Women_ = 4.70; *SD*_Women_ = 0.68; *M*_Men_ = 4.50; *SD*_Men_ = 0.79; *t*_417.19_ = 4.04; *p* < 0.001), and total attitude scores (*M*_Women_ = 4.52; *SD*_Women_ = 0.61; *M*_Men_ = 4.36; *SD*_Men_ = 0.70; *t*_422.32_ = 3.90; *p* < 0.001). The size effects ranged from 0.62 to 0.80, which indicate a moderate to high size effect, despite smaller mean differences. These are likely to be due to unequal variances between the groups, as the effect-size analysis can overestimate small mean differences when variances are not equal. In these cases, authors recommend observing Hedges’ *g*, which applies a correction for sample size and provides more reliable estimates [33], hence Hedges’ *g* indices were provided for all mean differences.

*T*-tests were replicated with the original ACO-25 questionnaire. Nurses presented higher affective attention scores (*M*_Student_ = 3.41; *SD*_Student_ = 0.32; *M*_Nurses_ = 3.69; *SD*_Nurses_ = 0.45; *t*_493.889_ = −11.41; *p <* 0.001), while students showed higher scores in behavioral (*M_Student_* = 4.55; *SD*_Student_ = 0.71; *M*_Nurses_ = 4.11; *SD*_Nurses_ = 0.83; *t*_521.535_ = 9.37; *p <* 0.001) and cognitive attitudes (*M*_Student_ = 4.72; *SD*_Student_ = 0.64; *M*_Nurses_ = 4.41; *SD*_Nurses_ = 0.88; *t*_473.160_ = 6.49; *p <* 0.001), as well as in the total score of the ACO-25 scale (*M*_Student_ = 3.71; *SD*_Student_ = 0.34; *M*_Nurses_ = 3.56; *SD*_Nurses_ = 0.52; *t*_481.752_ = 5.21; *p <* 0.001). The size effects for the differences between nursing students and nurses ranged from 0.35 to 0.73. As for gender differences, women scored higher than men in behavioral (*M*_Women_ = 4.51; *SD*_Women_ = 0.74; *M*_Men_ = 4.26; *SD*_Men_ = 0.79; *t*_442.119_ = 5.05; *p <* 0.001) and cognitive attitudes (*M*_Women_ = 4.70; *SD*_Women_ = 0.68; *M*_Men_ = 4.50; *SD*_Men_ = 0.79; *t*_417.191_ = 4.04; *p <* 0.001), as well as the total of the ACO-25 scale (*M*_Women_ = 3.71; *SD*_Women_ = 0.37; *M*_Men_ = 3.59; *SD*_Men_ = 0.40; *t*_438.077_ = 4.7; *p <* 0.001). The size effects, calculated through Hedges’ *g*, ranged from 0.36 to 0.75.

Intercorrelations between dimensions were calculated for both the abbreviated version (ACO-R) and the original version (ACO-25), for both students and nurses (Table 6). Correlation indices showed a low to moderate correlation with the affective dimension (for both behavioral and cognitive) and a moderate to high correlation between the behavioral and cognitive dimensions. These results were consistent across scales and for both students and nurses.

### 3.3. Comparison Criteria

Percentiles of ACO-R are provided (Table 7) by profile and gender; percentiles from 10 to 90 are offered for students, nurses, female students, female nurses, male students, and male nurses.

## 4. Conclusions

The present study aimed to develop a short form of a valid and reliable measure of attitudes toward patient communication in nurses and nursing students. For this purpose, three different objectives were formulated.

The first objective was to examine the psychometric properties of ACO-R, including its reliability, validity, and factorial structure. The internal consistency of the instrument was studied and showed optimal values in the subscales and in the total score, as well as in the two samples under study. The validity of the scale was assessed, which also proved to be positive, and a short-form version of the questionnaire (ACO-R) was proposed, consisting of 12 items divided equally into three subscales. The condensation of the instrument preserves the initial structure and offers a comparable solution in nursing students and nurses. As the literature suggests, developing a short form of a questionnaire is a preferred option when the project requires efficiency and participant engagement—as long as its psychometric properties are adequate [25,26]. Observing its sound reliability and validity indices, ACO-R can be considered a useful instrument for assessing attitudes towards communication in nurses and nursing students. As stated before, a short form may be beneficial to the study; it may increase motivation to respond and may be more economical in terms of time and resources spent on the assessment, both in professional practice and in research.

Analyses revealed the goodness-of-fit for a second-order factor, which adds nuance to the reliability and validity indices offered. The second-order factor model proposed in this study is grounded in the Theory of Reasoned Action [7] and the Theory of Planned Behavior [8], which conceptualize attitudes as multidimensional constructs comprising different but interrelated components. These theories suggest that attitudes are formed through the interplay of cognitive, affective, and behavioral factors, each contributing uniquely to the overall attitude. This proposal aligns with the current model, in which the global score represents an aggregation of three underlying dimensions that contribute equally to the overall score, reflecting a balanced integration of cognitive, emotional, and behavioral components. Specifically, the emotional dimension includes items that evaluate the stress experienced during communication, which is considered the defining element of this component. By framing the emotional dimension in this way, its unique contribution is acknowledged, while keeping the premise that all three dimensions are equally weighted in the total score.

Furthermore, measurement invariance was tested. Although full metric invariance was not established, the confirmation of partial metric invariance indicates that while some measurement parameters may vary across groups, a sufficient level of equivalence exists to ensure that the comparisons of the latent constructs remain meaningful and valid. This suggests that the constructs are interpreted similarly by nursing students and professionals for most of the items, enabling reliable group comparisons while acknowledging potential differences in specific measurement parameters.

The second objective was to analyze the mean differences across profile and gender. Students showed higher scores in both behavioral and cognitive domains, as well as in the overall attitudes score, when compared to nurses. These findings suggest a divergence in the perception of patient communication skills between these two groups. One possible reason for the discrepancy could be that students have been more recently immersed in academic learning, which often emphasizes theoretical and behavioral competencies, including empathy and communication skills. This focus may encourage more positive attitudes in the cognitive and behavioral domains. In contrast, nurses, who have more practical experience, might encounter real-world challenges—such as high patient loads, burnout, or institutional constraints—that could impact their overall attitudes, especially in domains related to patient interaction and adaptability in clinical settings [34,35]. Additionally, gender disparities were evident, with women consistently scoring higher than men across behavioral, cognitive, and total attitudes towards patient communication. These results imply potential gender-related differences in communication approaches within healthcare settings, adding to this line of research in the literature [13]. These results were replicated for ACO-25; however, size effects were higher in the short form of the questionnaire, revealing its strong psychometric properties compared to the original version.

The third objective aimed to provide comparison criteria for the different subgroups in the present sample. These data were provided for a wide sample of nurses and nursing students, serving as a valuable tool for future research. By using these comparison criteria, research on attitudes towards communication could identify relevant patterns across professional profile and gender, adding to this growing body of literature.

Despite the good results obtained, the present study has some limitations. First, the characteristics of the samples complicate the generalizability of the data to the general population, especially due to the sample size and the fact that the selection of participants was made by means of convenience sampling. It would therefore be interesting to evaluate nursing students and nurses from all over Spain. Moreover, the sample is composed mainly of women, so in the future, it would be desirable to have the responses of more men. Nevertheless, the study reflects the reality of the nursing profession, as most people in nursing are women [36]. Furthermore, the generalizability of the ACO-R questionnaire to nursing contexts outside of Spain should consider potential cultural differences in nurse–patient communication. Spanish nurses may prioritize interpersonal warmth and close relationships, reflecting a cultural emphasis on building personal rapport and attentiveness in patient care. Additionally, Spain’s healthcare model strongly involves family in patient care, potentially resulting in unique communication dynamics [37]. Therefore, cross-cultural validation studies would be beneficial to explore how these cultural nuances affect the questionnaire’s applicability and reliability across diverse healthcare settings, ensuring the robust interpretation of results. While this study highlights the utility of the ACO-R within the Spanish nursing context, future research should focus on validating the instrument across diverse cultural and clinical settings. Such studies would provide critical insights into how cultural and systemic differences influence nurse–patient communication, ensuring the questionnaire’s reliability and relevance in varied healthcare environments. These efforts are essential for extending the applicability of the ACO-R beyond Spain and to promote culturally competent communication practices within the nursing profession.

Additionally, future studies should aim to verify the convergent and discriminant validity of this version of the ACO-R by including relevant constructs, such as empathy or emotional intelligence measures. These constructs are theoretically relevant and could help establish the scale’s robustness and applicability across the different dimensions of interpersonal skills in nursing [20,21]. Therefore, the ACO-R instrument can be considered a useful, practical tool with which the attitudes of Spanish nurses towards communication can be evaluated.

Positive attitudes towards nurse–patient communication is a fundamental part of care from a biopsychosocial framework [6]. Knowing that the evaluation of personal relationships is a complex phenomenon, the development of tools that allow their evaluation with scientific rigor can contribute to a deeper understanding of how nursing students and nurses communicate with patients [1,18]. In this way, it can also help to plan future interventions to enhance positive attitudes towards communication among nurses [9,10]. This could have positive repercussions on a social, political, personal, economic, and healthcare scale [34].

Furthermore, these findings underscore the importance of considering professional status and gender when assessing attitudes towards patient communication. This study highlights potential areas for targeted interventions aimed at enhancing communication skills among healthcare professionals, as well as fostering more equitable communication practices across genders. Further research exploring the underlying factors contributing to these disparities could provide valuable insights for the development of specific communication training programs within healthcare education and practice.

This study could have important implications for both professionals and those beyond the nursing sector. Nurses can benefit from these insights into their communication skills and attitudes, informing targeted interventions and facilitating self-assessment. Researchers, on the other hand, could advance future research by using ACO-R, taking advantage of its solid psychometric properties and the efficacy and convenience of its use. Finally, policy makers and educators are urged to integrate communication training into healthcare education and professional development initiatives, tailoring programs to address specific needs identified within their contexts. This study could serve as guidelines for this purpose, providing valuable information not only about nursing students and nurses and their differences in communication, but also from a gender perspective.

In summary, the ACO-R for students and professional nurses has sufficient empirical support to be considered a valid and useful instrument with solid psychometric properties, and it works effectively in the evaluation of the perception of attitudes towards communication. Its implementation could help with advancing both individual professional development for nurses and the collective efficacy of nursing care, contributing to the overarching goal of optimizing patient well-being and healthcare quality.

## Figures and Tables

**Table 1 healthcare-12-02546-t001:** Descriptive statistics and reliability indices of all dimensions of ACO-R by profile.

	Students				Nurses			
	*M*	*SD*	r_jx_	α-x	*M*	*SD*	r_jx_	α-x
Affective	α = 0.90; AVE = 0.55; CR = 0.83				α = 0.76; AVE = 0.37; CR = 0.68			
ACO3	1.85	0.95	0.71	0.89	1.61	1.19	0.55	0.72
ACO5	1.51	0.82	0.78	0.87	1.62	1.13	0.48	0.75
ACO6	1.65	0.87	0.79	0.86	1.63	1.19	0.68	0.64
ACO7	1.59	0.87	0.81	0.85	1.58	1.09	0.56	0.71
Behavioral	α = 0.92; AVE = 0.55; CR= 0.83				α = 0.77; AVE = 0.43; CR = 0.75			
ACO14	4.58	0.84	0.78	0.90	4.25	1.08	0.56	0.73
ACO15	4.61	0.79	0.81	0.89	4.06	1.15	0.57	0.72
ACO16	4.66	0.75	0.86	0.87	4.23	1.01	0.65	0.68
ACO18	4.63	0.77	0.78	0.90	4.19	0.98	0.53	0.74
Cognitive	α = 0.94; AVE = 0.63; CR = 0.87				α = 0.85; AVE = 0.48; CR = 0.78			
ACO22	4.67	0.73	0.82	0.93	4.44	1.04	0.73	0.79
ACO23	4.75	0.66	0.87	0.92	4.40	1.03	0.75	0.78
ACO24	4.74	0.67	0.89	0.91	4.37	1.04	0.62	0.84
ACO25	4.76	0.69	0.85	0.92	4.43	1.07	0.66	0.82
Total	α = 0.91; AVE = 0.72; CR= 0.97				α = 0.89; AVE = 0.52; CR= 0.93			

Note: *M* = Mean; *SD* = Standard Deviation; α = Cronbach’s alpha; r_jx_ = item-total correlation; α-x = Cronbach’s alpha without the element; AVE = Average Variance Extracted; CR = Composite Reliability. For AVE, scores ≥ 0.50 were considered adequate; for α and CR, values ≥ 0.70 were considered adequate.

**Table 2 healthcare-12-02546-t002:** Confirmatory factor analysis indices.

	*χ* ^2^	*df*	S-B *χ*^2^	S-B *df*	RMSEA (90% CI)	IFI	CFI
ACO-R (3-factor)							
Students (*n* = 1408)	223.43 **	51	103.21 **	51	0.03 (0.020. 0.036)	0.98	0.98
Nurses (*n* = 385)	246.48 **	51	118.05 **	51	0.06 (0.048. 0.078)	0.91	0.91
ACO-R (2nd order)							
Students (*n* = 1408)	215.23 **	51	106.55 **	51	0.03 (0.020. 0.035)	0.98	0.98
Nurses (*n* = 385)	277.03 **	51	115.64 **	51	0.06 (0.044. 0.071)	0.93	0.93
ACO-25 (3-factor)							
Students (*n* = 1408)	2636.43 **	272	1392.93 **	272	0.05 (0.051. 0.057)	0.86	0.86
Nurses (*n* = 385)	1328.67 **	272	693.79 **	272	0.06 (0.058. 0.069)	0.82	0.82
ACO-25 (2nd order)							
Students (*n* = 1408)	2636.43 **	272	1392.93 **	272	0.05 (0.051. 0.057)	0.86	0.86
Nurses (*n* = 385)	1328.66 **	272	693.79 **	272	0.06 (0.058. 0.069)	0.82	0.82

Note: *χ*^2^ = chi-square statistic; *df* = degrees of freedom; S-B *χ*^2^ = Satorra–Bentler scaled chi-square; S-B *df* = degrees of freedom for the Satorra–Bentler scaled chi-square; RMSEA (90% CI) = Root Mean Square Error of Approximation with a 90% Confidence Interval; IFI = Incremental Fit Index; CFI = Comparative Fit Index. ** *p <* 0.01. For RMSEA, scores ≤ 0.08 were considered adequate. For CFI and IFI, values ≥ 0.90 were considered adequate.

**Table 3 healthcare-12-02546-t003:** Multigroup fit statistics for invariance models.

Single Group Solutions	*χ* ^2^	*Df*	∆*χ*^2^	∆*df*	*p*	RMSEA (90% CI)	IFI	CFI
Students (*n* = 1408)	215.23 **	51				0.03 (0.020, 0.035)	0.98	0.98
Nurses (*n* = 385)	277.03 **	51				0.06 (0.044, 0.071)	0.93	0.93
Measurement invariance								
Equal form	404.56	102				0.06 (0.05, 0.07)	0.98	0.98
Equal factor loading	493.82	111	89.26	9	<0.001	0.07 (0.06, 0.07)	0.97	0.97

Note: *χ*^2^ = chi-square statistic; *df* = degrees of freedom; ∆*χ*^2^ = difference in chi-square; ∆*df* = difference in degrees of freedom; *p* = *p*-value, significant at *p* < 0.05; ** *p <* 0.01. RMSEA (90% CI) = Root Mean Square Error of Approximation with a 90% confidence interval; CFI = Comparative Fit Index; IFI = Incremental Fit Index. For RMSEA and SRMR, scores ≤ 0.08 were considered adequate. For CFI and IFI, values ≥ 0.90 were considered adequate.

**Table 4 healthcare-12-02546-t004:** Parameter constraints for measurement invariance analysis.

Constraint	Corresponding Parameter	∆*χ^2^* (*df*)	*p*
22	(2, ACO6, F3) = 0.000	16.73 (1)	0.00
16	(1, ACO18, F3) = 0.000	12.12 (1)	0.00
4	(1, ACO15, F2) − (2, ACO15, F2) = 0	7.22 (1)	0.01
18	(1, ACO23, F2) = 0.000	6.11 (1)	0.01
10	(1, ACO5, F3) = 0.000	5.59 (1)	0.02
9	(1, ACO25, F3) − (2, ACO25, F3) = 0	2.86 (1)	0.09
6	(1, ACO18, F2) − (2, ACO18, F2) = 0	2.45 (1)	0.12
14	(1, ACO14, F3) = 0.000	1.99 (1)	0.16
21	(2, ACO7, F2) = 0.000	1.29 (1)	0.26
8	(1, ACO24, F3) − (2, ACO24, F3) = 0	1.06 (1)	0.30
27	(2, ACO23, F2) = 0.000	3.36 (1)	0.07
3	(1, ACO7, F1) − (2, ACO7, F1) = 0	0.59 (1)	0.44
13	(1, ACO14, F2) = 1.000	0.19 (1)	0.67
1	(1, ACO5, F1) − (2, ACO5, F1) = 0	0.15 (1)	0.70
23	(2, ACO7, F2) = 0.000	0.04 (1)	0.84
24	(2, ACO7, F3) = 0.000	0.56 (1)	0.46

Note: ∆*χ*^2^ = chi-square increment; *df* = degrees of freedom; *p* = *p*-value, significant at *p* < 0.05.

**Table 5 healthcare-12-02546-t005:** *t*-test scores by profile and gender.

		Students *M* (*SD*)	Nurses *M* (*SD*)	*t*	*p*	*g*	Women *M* (*SD*)	Men *M* (*SD*)	*t*	*p*	*g*
ACO-R	Affective	4.34 (0.77)	4.38 (0.90)	−0.77	0.22	0.80	4.37 (0.78)	4.29 (0.84)	1.62	0.05	0.80
	Behavioral	4.56 (0.74)	4.08 (0.90)	9.27	<0.001	0.78	4.51 (0.79)	4.27 (0.84)	4.56	<0.001	0.80
	Cognitive	4.72 (0.64)	4.41 (0.88)	6.49	<0.001	0.69	4.70 (0.68)	4.50 (0.79)	4.04	<0.001	0.70
	Total	4.54 (0.57)	4.29 (0.79)	5.74	<0.001	0.62	4.52 (0.61)	4.36 (0.70)	3.90	<0.001	0.62
ACO-25	Affective	3.41 (0.32)	3.69 (0.45)	−11.41	<0.001	0.35	3.47 (0.36)	3.47 (0.38)	0.05	0.48	0.36
	Behavioral	4.55 (0.71)	4.11 (0.83)	9.37	<0.001	0.73	4.51 (0.74)	4.26 (0.79)	5.05	<0.001	0.75
	Cognitive	4.72 (0.64)	4.41 (0.88)	6.49	<0.001	0.69	4.70 (0.68)	4.50 (0.79)	4.04	<0.001	0.70
	Total	3.71 (0.34)	3.56 (0.52)	5.21	<0.001	0.39	3.71 (0.37)	3.59 (0.40)	4.70	<0.001	0.38
	Min–Max	1–5					1–5				

Note: *M* = Mean; *SD* = Standard Deviation; *t* = *t*-value from the *t*-test, *p* = *p*-value indicating the significance level; *g* = Hedges’ *g* corrected effect size measure; Min–Max = minimum and maximum scores for the scale.

**Table 6 healthcare-12-02546-t006:** Intercorrelations between the dimensions for students and nurses in the ACO-R (abbreviated) and the ACO-25 (original).

	Students				Nurses			
	1	2	3	4	1	2	3	4
1. Affective	-	0.40 **	0.36 **	0.35 **	-	0.58 **	0.58 **	0.65 **
2. Behavioral	0.31 **	-	0.82 **	0.93 **	0.54 **	-	0.74 **	0.89 **
3. Cognitive	0.29 **	0.78 **	-	0.86 **	0.61 **	0.76 **	-	0.86 **
4. Total	0.69 **	0.86 **	0.84 **	-	0.84 **	0.88 **	0.90 **	-

Note: Results for ACO-R (below diagonal); ACO-25 (above diagonal). ** *p <* 0.01.

**Table 7 healthcare-12-02546-t007:** Cut-off scores of ACO-R by profile and gender.

PC	Students				Nurses					Female Students					Female Nurses					Male Students				Male Nurses			
	A	B	C	T		A	B	C	T		A	B	C	T		A	B	C	T	A	B	C	T	A	B	C	T
10	3.25	3.75	4.00	3.92		3.00	3.00	3.00	3.25		3.25	4.00	4.25	4.00		3.00	3.00	3.25	3.25	3.25	3.50	4.00	3.65	3.00	2.75	3.00	3.00
20	3.75	4.25	4.50	4.25		3.75	3.50	4.00	3.67		3.75	4.25	4.75	4.27		4.00	3.50	4.00	3.91	3.75	4.00	4.25	4.00	3.25	3.35	3.75	3.42
30	4.00	4.50	4.75	4.42		4.00	3.75	4.25	4.18		4.00	4.50	5.00	4.50		4.25	4.00	4.25	4.25	4.00	4.00	4.50	4.25	4.00	3.75	4.10	4.25
40	4.33	4.75	5.00	4.58		4.50	4.00	4.50	4.35		4.50	4.75	5.00	4.67		4.70	4.00	4.73	4.42	4.25	4.28	4.75	4.43	4.25	4.00	4.37	4.33
50	4.50	5.00	5.00	4.75		5.00	4.25	4.75	4.55		4.75	5.00	5.00	4.75		5.00	4.25	5.00	4.55	4.50	4.50	5.00	4.58	4.75	4.38	4.75	4.58
60	4.75	5.00	5.00	4.83		5.00	4.50	5.00	4.67		4.75	5.00	5.00	4.83		5.00	4.50	5.00	4.67	4.75	4.75	5.00	4.67	5.00	4.50	5.00	4.67
70	5.00	5.00	5.00	4.92		5.00	4.75	5.00	4.83		5.00	5.00	5.00	4.92		5.00	4.75	5.00	4.83	4.75	5.00	5.00	4.75	5.00	4.75	5.00	4.82
80	5.00	5.00	5.00	5.00		5.00	5.00	5.00	5.00		5.00	5.00	5.00	5.00		5.00	5.00	5.00	5.00	5.00	5.00	5.00	4.92	5.00	5.00	5.00	4.92
90	5.00	5.00	5.00	5.00		5.00	5.00	5.00	5.00		5.00	5.00	5.00	5.00		5.00	5.00	5.00	5.00	5.00	5.00	5.00	5.00	5.00	5.00	5.00	5.00

Note: PC = Percentile; A = Affective; B = Behavioral; C = Cognitive; T = Total of the scale.

## Data Availability

The database used in this work is available upon request to the corresponding author.

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
