# Peer review of "A Short Form of a Questionnaire on Attitudes Toward Patient Communication in Nurses and Nursing Students (ACO-R): Assessment of Psychometric Properties"

_healthcare, 2024, doi:10.3390/healthcare12242546_

Round 1
Reviewer 1 Report
Comments and Suggestions for Authors
The present manuscript describes the study of a brief form of the Attitudes toward Patient Communication questionnaire. The study is interesting and the sample size is sufficiently large. The manuscript has the potential to contribute to the knowledge in the field. However, my reading indicated several points that should be addressed before publication can be considered. Please find my comments below.
(1) Data availability statement is missing. For research transparency, data and syntaxes should be shared in an openly available repository such as the Open Science Framework.
(2) The introduction should provide more information about the reliability and validity (i.e., findings from factor analyses such as fit; external validity) of the measure. This would allow readers to derive an impression of the overall psychometric properties of the scale under investigation. Considering that this manuscript deals with the study of the ACO, it is surprising to only read "In previous studies, the instrument had shown adequate psy- 104
chometric properties in a sample of nurses and nursing students (Giménez- 105
Espert & Prado-Gascó, 2018; Giménez-Espert et al., 2021)." in the methods section.
(3) Research objectives should be clarified. Just stating that reliability and validity will be examined is unclear, as there are numerous ways to analyze them (e.g., internal consistency, retest-/split-half reliability; factorial validity, retrospective and prospective validity etc.).
(4) Please follow APA guidelines on the structure of manuscripts (e.g., participants should be described as first point under methods).
(5) Clarify who the experts are who judged the item content.
(6) Please provide measurement invariance analyses before conducting mean comparisons to clarify whether means can be compared in a meaningful way between groups, or whether mean differences are based on measurement-related differences.
(7) Authors examine a three-factor solution in the CFA, but later provide a total score without testing the assumption whether a higher-order factor exists. Please examine whether a second-order model and/or bifactor model explains the data by three factors and a general factor as suggested by the score computation.
(8) Check the d-value for the ACO-R Affective comparison. The values are unreasonably large (.80) considering that the t-test indicates no difference.
Author Response
Dear Editor,
First of all, we would like to thank you for your time and effort dedicated to our manuscript titled: “A short form of a questionnaire on Attitudes toward Patient Communication in nurses and nursing students (ACO-R): assessment of psychometric properties” submitted to Healthcare. We appreciate the suggestions offered by you and the reviewers; the letter below shows how we addressed them. New sections and changes to the body of the text are shown in blue. We believe the manuscript is now stronger and clearer. We will be happy, of course, to consider any further suggestions.
We are confident that the revisions have enhanced the clarity and strength of the manuscript. Thank you for your support.
On behalf of all the authors.
Yours sincerely,
Alicia Tamarit (Social Psychology Department, Faculty of Psychology and Speech Therapy, Universitat de València; alicia.tamarit@uv.es.
We would like to thank the reviewers for their kind words and suggestions. We have incorporated all the suggested changes into the text. We believe that the manuscript has significantly improved. Thank you for your valuable contributions.
Reviewer: 1
The present manuscript describes the study of a brief form of the Attitudes toward Patient Communication questionnaire. The study is interesting, and the sample size is sufficiently large. The manuscript has the potential to contribute to the knowledge in the field. However, my reading indicated several points that should be addressed before publication can be considered. Please find my comments below.
Reviewer: (1) Data availability statement is missing. For research transparency, data and syntaxes should be shared in an openly available repository such as the Open Science Framework.
Authors: Thank you for your feedback. We have included a Data Availability Statement as requested. Specifically, it reads: "The database used in this work is available on request to the corresponding author." (page 11).
Reviewer: (2) The introduction should provide more information about the reliability and validity (i.e., findings from factor analyses such as fit; external validity) of the measure. This would allow readers to derive an impression of the overall psychometric properties of the scale under investigation. Considering that this manuscript deals with the study of the ACO, it is surprising to only read "In previous studies, the instrument had shown adequate psy- 104. chometric properties in a sample of nurses and nursing students (Giménez- Espert & Prado-Gascó, 2018; Giménez-Espert et al., 2021)." in the methods section.
Authors: Additional information about the psychometric properties of the instrument has been included in the Introduction section. Specifically, details have been added on both the nursing and nursing students versions: The original 2018 study yielded adequate goodness-of-fit indices for a 25-item structure, for reliability indices (α = 0.76), EFA (RMSEA= 0.42; GFI =0.99) and CFA (S-Bχ2= 525.09; df = 272; p< 0.05; CFI=.91; IFI=.91) (Giménez-Espert & Prado-Gascó, 2018). These indices were also adequate in its 2021 version, adapted for nursing students, for reliability indices (α = 0.84), EFA ((RMSR= 0.03; GFI =0.99) and CFA (χ2 = 2347.59; df = 272; p< 0.05; CFI=.92; IFI=.92) (Giménez-Espert et al., 2021). (page 2)
Reviewer: (3) Research objectives should be clarified. Just stating that reliability and validity will be examined is unclear, as there are numerous ways to analyze them (e.g., internal consistency, retest-/split-half reliability; factorial validity, retrospective and prospective validity etc.).
Authors: The study objective has been modified to make it clearer. Specifically, “The objectives of this study were: (1) examine the psychometric properties of the ACO-R scale by evaluating its reliability through internal consistency, and assessing validity through factorial validity . (confirmatory factor analysis), and discriminant and convergent validity; (2) analyze mean differences in ACO-R scores across professional profiles (nursing students versus nurses) and gender (women versus men); and (3) provide comparative criteria for various subgroups within the sample. This comprehensive approach ensures a thorough psychometric evaluation and facilitates meaningful subgroup comparisons across nursing contexts.” (page 2).
Reviewer: (4) Please follow APA guidelines on the structure of manuscripts (e.g., participants should be described as first point under methods).
Authors: The participant information has been moved to the first section of the Methods, as requested. Additionally, the sections within the Methods and Results have been renumbered to ensure coherence throughout the document (pages 3-4).
Reviewer: (5) Clarify who the experts are who judged the item content.
Authors: Additional information has been provided regarding the validation process. Specifically, “A content validation process was conducted with more than five experts, as recommended in the literature (Sendjaya et al., 2008), specifically involving ten experts to ensure the clarity and relevance of the scale items. The panel included three experts in psychometrics, two psychologists, and five researchers in the field of nursing.” (page 5).
Reviewer: (6) Authors examine a three-factor solution in the CFA, but later provide a total score without testing the assumption whether a higher-order factor exists. Please examine whether a second-order model and/or bifactor model explains the data by three factors and a general factor as suggested by the score computation.
Authors: A second-order solution was examined and tested, yielding psychometrically sound results and adding richness to our results (page 5). This is explored in this manner: “Additionally, and as the score computation suggested, a second-order model was tested to examine whether a higher-order factor existed. Results showed that, for both students and nurses, a second-order model provided a marginally better fit for the data structure. This second-order model yielded a slightly higher χ², a slightly lower Satorra-Bentler χ², and, in the case of nurses, a slightly narrower confidence interval and higher IFI and CFI scores. These results indicate that a higher-order factor exists together with the 3-factor structure previously observed. Furthermore, indices for ACO-R were more adequate than those for ACO-25, suggesting the short form of the ACO questionnaire is psychometrically sound. This was true for both the 3-factor and second-order model calculated, which presented similar results.”
Reviewer: (7) Check the d-value for the ACO-R Affective comparison. The values are unreasonably large (.80) considering that the t-test indicates no difference.
Authors: Following the reviewer suggestion, we re-calculated size effect through Hedges’ g, which is a more accurate measure of size effect as it applies correction to the unequal variances (page 6). It is explained in the text: “Size effects ranged from .62 to .80, which indicate a moderate to high size effect, despite smaller mean differences. These are likely due to unequal variances between the groups, as effect-size analysis could overestimate small mean differences when variances are not equal. In these cases, au-thors recommend observing Hedges’ g, which applies a correction for sam-ple size and provides more reliable estimates (Goulet-Pelletier & Cousi-neau, 2018), hence Hedges’ g indices were provided for all mean differ-ences.”
Reviewer 2 Report
Comments and Suggestions for Authors
MS Title: A short form of a questionnaire on Attitude….
MS Authors: Tamarit et al.
MS Number: healthcare-3269213-
Date of Review: 2024/10/29
The authors applied a short-form questionnaire on attitudes (i.e., the Attitudes Toward Communication (ACO) scale: ACO-R) toward communication of nurses and nursing students to evaluate their patients’ health outcomes. Internal consistency, reliability, and validity of such short form of questionnaire were evaluated.
Comments:
1. Both the ACO scale with 25 items and short-form (ACO-R) with 12 items have been mentioned. It would be easier for the readers if both items can be presented in a table or put it in supplementary materials. Otherwise, it is difficult to justify.
2. The study subjects in this study include the nursing students and practicing nurses (lines 108-113). It would be informative if the years of the nursing students and the disciplines/working years of the practicing nurses be presented in each study group.
3. Again, 12 items (3, 5, 6, 7; 14, 15, 16, 18; and 22, 23, 24, 25) of the ACO were selected to construct the ACO-R (line-165). It is interesting to know how this selection was done by which standards or rationale.
4. Discussion section: “Students showed higher scores in both behavioral and cognitive domains, as well as in the overall attitudes score, when compared to nurses.” Any possible reason for this discrepancy?
5. Limitation section: Regarding the generalizability of these study results, are there any potential differences among the Spanish nursing students and practicing nurses? And, how different the Spanish nurses could be from those in EU or in the western world (i.e., issues of nurse-patient communication)? Generalizability means the applicability of this short-form questionnaire.
6. As a hypothetic question, in these 1681 Spanish nursing students and practicing nurses, will their empathy quotient or emotional Intelligence quotient be relevant to the results obtained by the ACO or ACO-R?

Author Response
Dear Editor,
First of all, we would like to thank you for your time and effort dedicated to our manuscript titled: “A short form of a questionnaire on Attitudes toward Patient Communication in nurses and nursing students (ACO-R): assessment of psychometric properties” submitted to Healthcare. We appreciate the suggestions offered by you and the reviewers; the letter below shows how we addressed them. New sections and changes to the body of the text are shown in blue. We believe the manuscript is now stronger and clearer. We will be happy, of course, to consider any further suggestions.
We are confident that the revisions have enhanced the clarity and strength of the manuscript. Thank you for your support.
On behalf of all the authors.
Yours sincerely,
Alicia Tamarit (Social Psychology Department, Faculty of Psychology and Speech Therapy, Universitat de València; alicia.tamarit@uv.es.
We would like to thank the reviewers for their kind words and suggestions. We have incorporated all the suggested changes into the text. We believe that the manuscript has significantly improved. Thank you for your valuable contributions.
Reviewer: 2
The authors applied a short-form questionnaire on attitudes (i.e., the Attitudes Toward Communication (ACO) scale: ACO-R) toward communication of nurses and nursing students to evaluate their patients’ health outcomes. Internal consistency, reliability, and validity of such short form of questionnaire were evaluated.
Reviewer: Both the ACO scale with 25 items and short-form (ACO-R) with 12 items have been mentioned. It would be easier for the readers if both items can be presented in a table or put it in supplementary materials. Otherwise, it is difficult to justify.
Authors: We appreciate the reviewer’s suggestion. However, we regret that we cannot provide this information explicitly in the manuscript, as the instruments are patented (registration number: UV-MET-201917R).
Reviewer: The study subjects in this study include the nursing students and practicing nurses (lines 108-113). It would be informative if the years of the nursing students and the disciplines/working years of the practicing nurses be presented in each study group.
Authors: Their years of experience and disciplines have been included (page3): “Their employment situation was divided proportionally between tempo-rary workers, in which 56 (14.50%) had a temporary contract and 102 (26.50%) are temporary workers, and those with a permanent position, which were 185 (48.10%) nurses. Their years of experience ranged from 1 month to over 43 years (M= 21.64; SD = 20.50). They worked mainly in the Emergency department (n=41, 10.6%), Intensive Care Unit (ICU) (n=37, 9.6%), Surgery (n=28, 7.3%) and Internal Medicine (n=26, 6.8%), but they also worked in many other sections, such as Digestive, Pediatrics, Neona-tology, Hemodialysis, Trauma, Pulmonology, Intensive Respiratory Reha-bilitation Unit (REA), Operating Room, Cardiology, Comprehensive Medi-cine, and Oncology-Hematology-Cardiology, among others.”
Reviewer: Again, 12 items (3, 5, 6, 7; 14, 15, 16, 18; and 22, 23, 24, 25) of the ACO were selected to construct the ACO-R (line-165). It is interesting to know how this selection was done by which standards or rationale.
Authors: Information has been added, specifically, “A content validation process was conducted with more than five experts, as recommended in the literature (Sendjaya et al., 2008), specifically involving ten experts to ensure the clarity and relevance of the scale items. The panel included three experts in psychometrics, two psychologists, and five researchers in the field of nursing.” (page 5).
Reviewer: Discussion section: “Students showed higher scores in both behavioral and cognitive domains, as well as in the overall attitudes score, when compared to nurses.” Any possible reason for this discrepancy?.
Authors: A possible explanation for this discrepancy has been added. Specifically, we have proposed that “one possible reason for the discrepancy could be that students are more recently immersed in academic learning, which often emphasizes theoretical and behavioral competencies, including empathy and communication skills. This focus may encourage more positive attitudes in the cognitive and behavioral domains. In contrast, nurses, who have more practical experience, might encounter real-world challenges—such as high patient loads, burnout, or institutional constraints—that could impact their overall attitudes, especially in domains related to patient interaction and adaptability in clinical settings. (page 9)”
Furthermore, the following references have been included to support this argument:
Foster, K., Roche, M., Delgado, C., Cuzzillo, C., Giandinoto, J. A., & Furness, T. (2018). Resilience and mental health nursing: An integrative review of international literature. International Journal of Mental Health Nursing, 28(1), 71-85. https://doi.org/10.1111/inm.12548
Montgomery, A., Panagopoulou, E., Esmail, A., Richards, T., & Maslach, C. (2021). Burnout in healthcare: The case for organisational change. BMJ, 372, n52. https://doi.org/10.1136/bmj.n52
Reviewer: Limitation section: Regarding the generalizability of these study results, are there any potential differences among the Spanish nursing students and practicing nurses? And, how different the Spanish nurses could be from those in EU or in the western world (i.e., issues of nurse-patient communication)? Generalizability means the applicability of this short-form questionnaire.
Authors: Information regarding the generalizability of the data has been included. Specifically, we have added details on potential cultural factors that may impact nurse-patient communication in Spain compared to other EU or Western contexts. “The generalizability of the ACO-R questionnaire to nursing contexts outside of Spain should consider potential cultural differences in nurse-patient communication. Spanish nurses may prioritize interpersonal warmth and close relationships, reflecting a cultural emphasis on building personal rapport and attentiveness in patient care. Additionally, Spain’s healthcare model strongly involves family in patient care, potentially shaping unique communication dynamics (De-María et al., 2024). Therefore, cross-cultural validation studies would be beneficial to explore how these cultural nuances affect the questionnaire’s applicability and reliability across diverse healthcare settings, ensuring robust interpretation of results” (page 10).
Additionally, a supporting reference has been provided to substantiate this information.
De-María, B., Topa, G., & López-González, M. A. (2024). Cultural competence interventions in European healthcare: A scoping review. Healthcare, 12(10), Article 1040. https://doi.org/10.3390/healthcare12101040
Reviewer: As a hypothetic question, in these 1681 Spanish nursing students and practicing nurses, will their empathy quotient or emotional Intelligence quotient be relevant to the results obtained by the ACO or ACO-R?.
Authors: Thank you for the suggestion. Information has been added to the section on future research directions. Specifically, we included: "Additionally, future studies should aim to verify the convergent and discriminant validity of this version of the ACO-R by including related constructs, such as empathy or emotional intelligence measures. These constructs are theoretically relevant and could help establish the scale's robustness and applicability across different dimensions of interpersonal skills in nursing (Giménez-Espert et al., 2023; Sanchis et al., 2023)." (page 10).
Reviewer 3 Report
Comments and Suggestions for Authors The article is highly relevant to the training and professional development of nursing professionals. It addresses a critical competency within the profession, emphasizing how healthcare professionals' attitudes can either promote or hinder the development and application of this competence.The use of the proposed scale represents a valuable advancement in current nursing knowledge, offering a new tool for assessing this essential aspect of professional practice. The article is well-written and structured, building effectively on previous research in the field.
However, there are a few areas where improvements could be made. First, it would be beneficial to provide a more specific definition of the sample population, clarifying the characteristics and selection criteria of the participants. Additionally, more details on the methodology should be included, particularly regarding how the questionnaire was administered to the two groups of participants and the procedures for data collection. Finally, it is important to ensure that the study clearly addresses how anonymity and confidentiality were maintained throughout the research process.
Author Response
Dear Editor,
First of all, we would like to thank you for your time and effort dedicated to our manuscript titled: “A short form of a questionnaire on Attitudes toward Patient Communication in nurses and nursing students (ACO-R): assessment of psychometric properties” submitted to Healthcare. We appreciate the suggestions offered by you and the reviewers; the letter below shows how we addressed them. New sections and changes to the body of the text are shown in blue. We believe the manuscript is now stronger and clearer. We will be happy, of course, to consider any further suggestions.
We are confident that the revisions have enhanced the clarity and strength of the manuscript. Thank you for your support.
On behalf of all the authors.
Yours sincerely,
Alicia Tamarit (Social Psychology Department, Faculty of Psychology and Speech Therapy, Universitat de València; alicia.tamarit@uv.es.
We would like to thank the reviewers for their kind words and suggestions. We have incorporated all the suggested changes into the text. We believe that the manuscript has significantly improved. Thank you for your valuable contributions.
Reviewer: 3
Reviewer: The article is highly relevant to the training and professional development of nursing professionals. It addresses a critical competency within the profession, emphasizing how healthcare professionals' attitudes can either promote or hinder the development and application of this competence.
The use of the proposed scale represents a valuable advancement in current nursing knowledge, offering a new tool for assessing this essential aspect of professional practice. The article is well-written and structured, building effectively on previous research in the field.
However, there are a few areas where improvements could be made. First, it would be beneficial to provide a more specific definition of the sample population, clarifying the characteristics and selection criteria of the participants. Additionally, more details on the methodology should be included, particularly regarding how the questionnaire was administered to the two groups of participants and the procedures for data collection. Finally, it is important to ensure that the study clearly addresses how anonymity and confidentiality were maintained throughout the research process.
Authors: Firstly, we appreciate the reviewer’s positive feedback. Additional information has been included to clarify these aspects, specifically addressing how anonymity and confidentiality were maintained throughout the study. Specifically, “The inclusion criteria for students were that participants were actively en-rolled in an official nursing degree, and having consented their anonymous and confidential participation.
In Spain, the nursing degree program spans four academic years, which totals 240 credits under the European Credit Transfer System (ECTS), with students completing 60 ECTS credits per year. For scale, each ECTS credit corresponds to approximately 30 hours of academic work. Once this program finishes, graduates receive a nursing degree that quali-fies them to practice professionally in Spain as well as in other countries within the European Union.
As for the sample of nurses, it was collected from six hospitals in the Valencian Community (Spain). The inclusion criteria for nurses were that they were actively employed, whether by temporary or permanent con-tracts, and having consented their anonymous and confidential participa-tion. Therefore, the study contemplated a single evaluation through a cross-sectional design.
All participants signed the informed consent. This study was author-ized by their respective ECCR (Ethics Committees for Clinical Research) and the Research Ethics Comittee of the University where the study was performed (H1529396558647). After this, participants were given three weeks to complete the questionnaire, which lasted around 10 minutes, and were provided with a collection box at each unit to deposit their confiden-tial answers. These collection boxes were provided to ensure the answers were anonymous and to prevent any possible identification of the partici-pant. In order to improve response rates, reminder emails were sent two weeks after the initial distribution, and completed questionnaires were re-trieved after three weeks.” (page 3).